# Morphological Diversity of Permanent Maxillary Lateral Incisors and Their Impact on Aesthetics and Function in Orthodontically Treated Patients

**DOI:** 10.3390/diagnostics12112759

**Published:** 2022-11-11

**Authors:** Anita Fekonja

**Affiliations:** 1Department of Orthodontics, Healthcare Centre Maribor, Ulica Talcev 9, 2000 Maribor, Slovenia; anita.fekonja1@guest.arnes.si; 2Faculty of Medicine, University of Maribor, Taborska ulica 8, 2000 Maribor, Slovenia

**Keywords:** permanent maxillary lateral incisor, crown shape, developmental anomalies, aesthetics, function

## Abstract

The aim of this study was to determine the frequency of different crown shapes and associated dental anomalies of the permanent maxillary lateral incisor (PMLI) and their impact on aesthetics and occlusion (function) in orthodontically treated patients. Materials and Methods: The records of 372 subjects, which consisting of study casts, panoramic radiographs and anamnestic data, were investigated for crown shape and associated dental anomalies of PMLI and potential dental treatment to achieve satisfactory aesthetics and function. Descriptive statistics, including means, standard deviations and percentages for the observed variables, were calculated. Data were analyzed using the chi-square test. *p*-Values < 0.05 were considered as statistically significant. Results: The results showed that the most common crown shapes of PMLI were trapezoidal-shaped (59.8%), followed by central incisor-shaped (26.7%), canine-shaped (11.2%) and peg-shaped (2.3%), without statistically significant difference between genders. Developmental anomalies were found in 86 (11.6%) PMLI. All subjects with developmental anomalies were included in orthodontic treatment, and 91.2% of them need interdisciplinary treatment to achieve satisfactory aesthetics and function. Conclusions: The morphological diversity and developmental anomalies of the PMLI may affect aesthetics and function and should be considered in treatment planning.

## 1. Introduction

The permanent maxillary lateral incisors (PMLI) are paired maxillary teeth located distally from both permanent maxillary central incisors and medially from both permanent maxillary canines and thus have an impact on aesthetics, as well as function [1,2]. The number, size and shape of teeth are determined in the initial and morphogenetic phases of odontogenesis [2,3]. The first sign of calcification of the PMLI is visible at 10–12 months, and the crown is completed at 4–5 years of age. The tooth usually erupts into the oral cavity between 7 and 9 years of age, replacing the deciduous maxillary lateral incisor. The root continues to mineralize and its completion occurs at age 11 years [1,2]. The PMLI, similar to all the incisors, have the function of shearing or cutting food. They also support the upper lip and maintain it in its correct position and participate in voice formation (speech) [1,2].

The PMLI crown has the greatest degree of variation In shape, except for the third molars. Its general crown shape is similar to the permanent maxillary central incisor with a smaller and slightly more rounded crown or shows different crown shapes, such as canine-shaped or peg-shaped. Some dental developmental anomalies, for example, palatal pit, palatocervical groove, dens invaginatus, talon cusp and tooth agenesis, are also more commonly present in PMLI [1,2]. In PMLI, the mesial and distal marginal ridges, as well as the cingulum, are relatively more prominent; thus, the palatal fossa is deeper when compared to the maxillary central incisor. A palatocervical groove is more common in PMLI than in the central incisors. A palatocervical groove usually originates in the palatal pit and extends cervically to the root surface. The palatal pit and palatocervical groove can be considered a clinically unfavorable factor due to the plaque accumulation, which may lead to periodontal problems [2]. Dens invaginatus is a dental developmental anomaly characterized by an invagination of the enamel organ into dental papilla before calcification is completed and seen as a pit or a fissure on the oral surfaces [4,5]. The invagination begins in the crown and may extend into the root, and according to severity, various classifications have been proposed [6]. The teeth affected with this anomaly are often without clinical problems (asymptomatic) and can be detected during clinical and radiographic examination. The talon cusp is a form of dens evaginatus and presents as an accessory cusp-like structure usually found on the incisal part of the cingulum area or cementoenamel junction [7]. The talon cusps might cause clinical problems related to increased caries susceptibility, occlusal interferences and complication during endodontic, surgical and periodontal treatment [8].

Different crown shapes (for example peg-shaped) and developmental anomalies can affect the aesthetic appearance and function of teeth and can sometimes cause dental problems. Many factors must be taken into consideration during treatment, which depends on the patient’s expectations and the expertise of the clinician. The type of the treatment should be selected based on the aesthetic and functional requirements [9].

The etiology of dental developmental anomalies is multifactorial, with the interaction of genetic and environmental factors [10,11,12,13]. Understanding the process of dental morphogenesis and the resulting variations in the shape of the tooth crown, as well as the developmental abnormalities present, contributes significantly to a multidisciplinary approach to achieve satisfactory aesthetics and function. The PMLI plays an important role in the smile aesthetics, as well as facial aesthetics and individual’s general perception of life [14,15].

The crown of PMLI vary in shape more than any other permanent teeth, except the permanent third molars; therefore, the objective of this study was to investigate the prevalence of different crown shapes of the PMLI and the prevalence of PMLI-related dental developmental anomalies and to determine their impact on aesthetics and function in orthodontically treated patients in Slovenia.

## 2. Materials and Methods

The study was conducted in accordance with the Declaration of Helsinki at the Orthodontic Department of Community Healthcare Centre. Ethical approval for the study was obtained from the Institutional Review Board (No. 19/11). Informed consent approval was obtained from each patients’ parents or patients. The consent included their approval to use and publicize their data for educational and scientific research purposes.

In this study, the dental records of 698 subjects were reviewed. All records included the subjects’ data, with study casts, panoramic radiographs and anamnestic data (dental histories) before orthodontic treatment. Inclusion criteria for the study were subjects with fully erupted all permanent incisors (subjects at the end of early mixed dentition), high-quality study casts that allow the assessment of PMLI morphology and panoramic radiographs that fulfilled the inclusion criteria of the standard images of good quality, without any degree of exposure or positioning errors. Exclusion criteria were oral cavity-related syndromes, the alveolar cleft and/or palate, as well as fillings or a previous loss of PMLI due to trauma, caries or periodontal disease.

Of these 698 subjects, 294 subjects were excluded because they were in their early mixed dentition (all maxillary incisors were not erupted); 17 subjects were excluded due to large fillings of PMLI; 8 subjects were excluded due to the poor quality of study casts; 5 subjects were excluded due to fractured maxillary lateral incisors; 1 subject was excluded due to Down syndrome and 1 subject was excluded due to the present cleft lip, alveolar bone and palate; thus, the total number of patients was 372.

All subjects’ records were reviewed by a single examiner (AF). The study cast and panoramic radiographs were observed to assess the crown morphology of PMLI and associated dental anomalies. A tooth was registered as congenitally missing (hypodontia) when no sign of the tooth crown mineralization could be found on panoramic radiographs, and the data records and anamnestic data confirmed that the tooth had not been lost due to trauma, caries or other reasons.

The crown morphology was evaluated and classified as trapezoidal-shaped, central incisor-shaped, canine-shaped and peg-shaped (Figure 1).

A trapezoidal-shaped permanent maxillary lateral incisor (Figure 1a) is smaller in size (shorter and narrower) than the permanent maxillary central incisor, the labial surface is more convex in form compared to the central incisor and the distal outline is more rounded with contact point at the center of the middle third.

A central incisor-shaped permanent maxillary lateral incisor (Figure 1b) is smaller in size than the central incisor but has the same general appearance.

A canine-shaped permanent maxillary lateral incisor (Figure 1c) has an incisal edge divided with a mild cusp tip.

A peg-shaped permanent maxillary lateral incisor (Figure 1d) has a small conical crown.

Evaluated associated dental anomalies:

Hypodontia (Figure 2a): the permanent maxillary lateral incisor is congenitally missing.

Deep palatal fossa with palatal pit (Figure 2b): When the cingulum is well-prominent and associated with prominent marginal ridges, the palatal fossa is deep and palatal pit is located on the incisal surface of the cingulum.

Dens invaginatus (Figure 2c): an invagination of enamel and dentine diagnosed radiologically.

Palatocervical groove (Figure 2d): The groove originates in the palatal pit and extends cervically and slightly distally on to the cingulum or may extend up on the root.

Talon cusp (Figure 2e): is an accessory cusp on the palatal surface (usually on a cingulum area or cementoenamel junction).

The morphological type of PMLI and associated dental anomalies (hypodontia, talon cusp, palatal pit, dens invaginatus and palatocervical groove) in a single subject, as well as treatment that has been carried out, were recorded. Sex was also observed.

### Statistical Analysis

Statistical Package for Social Science (SPSS version 10.0; Chicago, IL, USA) was used for the statistical analysis. Means, standard deviations (SD) and percentages were computed for descriptive variables. The chi-square test was used to analyze differences in the distribution of different crown shapes and associated anomalies by sex. *p*-Values < 0.05 were regarded as statistically significant.

## 3. Results

The study group included 372 subjects (126 (33.9%) males and 246 (66.1%) females) with a mean age 10.71 ± 2.06 years (10.42 years ± 2.01 in males and 10.91 years ± 2.07 in females). There was no statistically significant difference in age (*p* > 0.05) between the sexes. Out of a total of 372 subjects, the hypodontia of PMLI was diagnosed in 15 (4.03%) subjects. There was a total of 21 congenitally missing PMLI. In others, 723 PMLI crown morphology and associated dental anomalies were investigated.

The distribution of various morphological shapes of PMLI among the sexes is shown in Table 1. Trapezoidal-shaped was the most common crown shape of PMLI (59.8%), followed by central incisor-shaped (26.7%), canine-shaped (11.2%) and peg-shaped (2.3%). No statistically significant differences between males and females (*p* > 0.05) were noted. In total of 13 (3.5%) subjects with peg-shaped PMLI, in 9 (69.2%) subjects, it was unilateral (5 on left side and 4 on right side) and, in 4 (30.8%) subjects, bilateral.

All 13 subjects with peg-shaped PMLI required additional treatment to achieve satisfactory aesthetics and function. Ten (76.9%) of them were treated conservatively by reshaping the tooth crown with composite by a restorative dentist (Figure 3), two (15.4%) subjects were treated with veneers by prosthodontist, and in one (7.7%) subject, the PMLI was extracted by an oral surgeon, space closed orthodontically and canine reshaped in PMLI. 11 (84.6%) subjects treated by restorative dentist, and all subjects treated with veneers by the prosthodontist and oral surgeon were satisfied with the treatment. Two subjects treated by a restorative dentist were dissatisfied with the treatment results and required retreatment.

Dental developmental anomalies were found in 86 (11.9%) PMLI of 57 (23.1%) subjects. The prevalence of various dental developmental anomalies of PMLI among males and females are shown in Table 2.

The most common developmental anomaly was deep palatal pit (6.5%), followed by hypodontia (4.03%), palatocervical groove (2.4%), talon cusps (1.3%) and dens invaginatus (1.1%), without statistically significant difference between the sexes (*p* > 0.05).

The overall prevalence of PMLI hypodontia was 4.03%. In 15 subjects (7 males and 8 females), 21 permanent maxillary lateral incisors were affected with hypodontia; in 6 subjects, bilaterally; and in 9 subjects, unilaterally. No statistically significant differences between males and females and unilateral and bilateral hypodontia were noted (*p* > 0.05). All patients with hypodontia required additional treatment to achieve satisfactory aesthetics and function. The treatment option showed Table 3.

In 24 (6.5%) subjects (9 males and 15 females), we found well-prominent cingulum and both marginal ridges with deep palatal fossa and palatal pit located on the incisal surface of the cingulum. In 18 subjects (5 males and 13 females), the palatal pit was found bilaterally and, in six subjects (four males and two females), unilaterally. No statistically significant differences between males and females (*p* > 0.05) were noted. In 22 subjects, the palatal pits were treated by a dentist with the filling of pits to prevent plaque retention and periodontal problems. In two subjects were diagnosed caries with affected pulp, and additional endodontic treatment was required.

In the present study, talon cusp was found in five subjects (1.3%), on seven PMLI, without statistically significant differences between males and females and unilateral and bilateral side (*p* > 0.05). In all five (100%) subjects, talon cusp hindered the correct placement of the PMLI (Figure 4) and occlusion, so it was necessary to remove it. All seven talon cusps have an enamel layer covering a dentin without pulp; therefore, only removal of the tubercula and polishing of the surface was possible without endodontic treatment.

Dens invaginatus was found in four (1.1%) samples (one male and three females) unilaterally and in all cases classified as type II according to the Oehler’s classification [6]. Two dens invaginatus were asymptomatic and diagnosed accidentally on radiographs during routine orthodontic examinations. They were treated with preventive filling of palatal pits to prevent plaque deposits. Two dens invaginatus had periodontal problems and was endodontically treated with included apexification (Figure 5).

A cervicopalatal groove was found in nine (2.4%) subjects (three males and six females). In three subjects (one male and two females), a cervicopalatal groove was found bilaterally and, in six subjects (two males and four females), unilaterally. There were no statistically significant differences between males and females and the left and right sides (*p* > 0.05). The groove was treated by a periodontologist in terms of protection to prevent plaque retention and periodontal problems.

All subjects were included in orthodontical treatment. In the present study, we found a statistically significant difference between subjects with trapezoidal-shaped, central incisor-shaped and canine-shaped PMLI and subjects with peg-shaped PMLI according to involvement with other specialists (*p* < 0.05). Additionally, 91.2% of subjects with dental developmental anomalies had an additional treatment by different dental specialists to achieve satisfactory aesthetics and function (Table 3).

## 4. Discussion

The PMLI is the tooth with the most variable crown shape, right after the third molar. Abnormalities in shape and size and also in number of PMLI is not a rarity in any orthodontical practice; yet, the treatment for this condition is usually a challenge due to affected aesthetic and function. They are an important clinical and public health problems, too [8,16].

Women showed more rounder, softer and more delicate tooth form in contrast men are perceived as angular and square that harmonize with their appearance [1]. In the present study the most common crown shape of PMLI was trapezoidal-shaped (59.8%), followed by central incisor-shaped (26.7%), canine-shaped (11.2%) and peg-shaped (2.3%) which is different from study by Schlegel and Satravaha [17].

The studies by Backman and Wahlin [18], Ling et al. [19] and Hua et al. [20] reported that the overall prevalence of peg-shaped maxillary permanent lateral incisor was 0.8%, 1% and 1.8%, respectively. The prevalence in our study was higher (2.3%) due to orthodontic samples. Hua et al. [20] reported that prevalence of peg-shaped PMLI varies by race, population and sex. The prevalence rates were higher among Mongoloid people, orthodontic patients and women. The prevalence rates were higher in orthodontic patients (2.7%) than in the general population (1.6%) and dental patients (1.9%) which is in agreement with our study. They also reported that prevalence rates of unilateral and bilateral peg-shaped incisors were approximately the same. However, among the unilateral PMLI, the left side was twice as common as the right side and in addition, contralateral PMLI hypodontia was seen in 55.5% of the subjects with unilateral PMLI. In contrast, Ijaz et al. [21] reported that peg-shaped PMLI was found more common bilateral (53.7%), followed by right side (24.4%) and left side (22.0%). In present study, the prevalence of peg-shaped PMLI was higher in comparison with previous studies [17,18,19,20]. We also found higher rates of unilateral (69.2%) than bilateral (30.8%) peg-shaped incisors, and among the unilateral side, left PMLI was more common affected (52.9%). In present study was found 5 (55.6%) subjects with unilateral peg-shaped PMLI and contralateral hypodontia of PMLI what is in agreement with study by Hua et al. [20]. In present study, statistically significant more subjects with peg-shaped PMLI needed additional dental specialist to achieved satisfactory aesthetics and occlusion than subjects with other crown shape. Unilateral peg-shaped PMLI and even more often unilateral hypodontia, may cause midline asymmetry which together with space (gaps) leads to an unattractive smile (Figure 6). The treatment generally requires an interdisciplinary approach which mean a combination of the orthodontic managements and restorative dentist or prosthodontist. Important factors in determining treatment planning are the age of the patient, the size of the crown and space. There are two basic approaches to treat peg-shaped PMLI. First, to open the space orthodontically for a normal-sized PMLI and then restorative dentist build up the PMLI to simulate a normal-sized PMLI. The second option is to extract PMLI and the resultant space close. In present study, in 12 (92.3%) subjects PMLI were treated conservatively to open space and simulate a normal PMLI, and in one (7.7%) subject the PMLI were extracted, space closed orthodontically and canines reshaped in PMLI.

Several previous studies reported that the PMLI are the third most frequent developmentally absent teeth after third molars and mandibular second premolars [22,23,24,25]. Approximately 2% of the UK population have developmentally absent PMLI [26]. The prevalence is higher in Europe and Australia than in North America, females are affected about 1.34 times more than males and bilateral absence is more common than the unilateral [25]. In present study, hypodontia of PMLI was found in 15 (4.03%) patients, with higher prevalence in females than in males, but without statistically significant difference (*p* > 0.05), which is consistent with the results of several previous authors [25,26]. The higher prevalence of hypodontia might be due to orthodontical samples used in this study. Unilateral hypodontia (60%) was more common than bilateral (40%). As previous mentioned unilateral peg-shaped PMLI is frequently associated with hypodontia of contralateral PMLI. In the present study, unilateral hypodontia of PMLI was found to be associated with peg-sharped contralateral incisor in 5 (55.5%) samples what is in agreement with study by Hua et al. [20]. An early diagnosis of hypodontia is very important for preventing aesthetic, functional and psychological changes. In subjects with hypodontia the treatment is complex and for successful and satisfied results requires an interdisciplinary approach including the collaboration between specialist, mainly with an orthodontist, a prosthodontist, an oral surgeon, and occasionally also with periodontist. As in previously described peg-shaped PMLI, the treatment plan depends on the age of the patient, the space, the condition of supporting tissues, occlusion and interocclusal space, and there are two treatment approaches possible: the first is closure of the space via orthodontic management [27] and the second is the replacement of the absent teeth via implants or prosthetics (bridge) [28]. In the present study, all subjects with hypodontia were treated interdisciplinarily. In 10 (66.7%) subjects, the space was closed, and the canines were reshaped conservatively or by veneers. The other five (33.3%) subjects were treated by maintain of the space for implant. A dental implant can be inserted after the end of the growth of the jawbone. Until this period, the space was maintained with a retainer on which a tooth is added for aesthetic reasons.

Dental developmental anomalies like talon’s cusp, dens invaginatus, and palatocervical groove are fairly seen in PMLI and can lead to various clinical problems like attrition, compromised aesthetics, displacement (misalignment) of the involved teeth, occlusal interference, increased susceptibility to dental caries and occlusal disharmony [29,30].

In the literature, we found that the talon cusp is usually found on the oral surface of maxillary incisors [31]. The incidence of talon cusps ranges from 0.2 to 5.2% and can be an isolated condition what is more common, or they can be associated with syndromes such as Rubenstein-Taybi [32,33]. In the present study, the incidence of talon cusps was 1.3% what is in agreement with the previous study. The talon cusp presented in this study extended from the cementoenamel junction to 0.5 mm short of the incisal edge, which is a type 1 [34]. Talon cusp raises aesthetic, functional and pathological challenges. Various treatment modalities have been followed for the management of talon cusps according to the type of the presentation and complication of talon cusps. Small and usually asymptomatic talon cusps require no therapy (treatment), while treatment of large talon cusps depends on the pulpal extension. The occlusal interference caused by large talon cusp include the reducing the cusp gradually and periodically with the application of topical fluoride or a total reduction of the cusp and calcium hydroxide pulpotomy [35]. Orthodontic treatment can be also involved due to tooth displacement or malalignment [36]. In this study, in all sample the talon cusp was gradually reduced and continuously protected with application of topical fluoride and finally with orthodontic appliance aligned to achieve satisfied function and aesthetics.

Some previous studies reported the prevalence of dens invaginatus of 0.04–10% and more common in male [4,31] and PMLI are the most commonly affected teeth with dens invaginatus [37]. In the present study the prevalence of dens invaginatus was 0.84%, what is in agreement with study by Hamasha and Alomari [37] but much lower that prevalence reported by Alkadi et al. [38] who reported 3.81% prevalence of dens invaginatus in PMLI. The difference may also be the result of different detection, since Alkadi et al. [38] used cone-beam computed tomography (CBCT) for detection dens invaginatus which is more accurate than panoramic radiograph used in present study. In the present study, dens invaginatus were in all samples found unilaterally, while Hamasha and Alomari [37] and Alkadi et al. [38] observed 30.1% and 24% dens invaginatus bilaterally, respectively. In the present study all dens invaginatus were type II, based on the Oehler’s classification [6]. Alkadi et al. [38] reported that type I was the most frequently observed (80%), followed by type II (17.8%) and type III (2.2%). Dens invaginatus is often asymptomatic and if undiagnosed, affected teeth are prone to developing caries and periapical pathology. Therefore, if PMLI shows any clinical features of dens invaginatus like deep palatal pits, additional radiographic examination should be performed to exclude the possibility of an underlying dens invagination [39]. The anatomy of dens invaginatus can be extremely complex and consequently the treatment can range from restorative to extraction of the tooth therefore early diagnosis and preventive treatment is important to prevent pulpal pathology [40,41]. In the present study two dens invaginatus were asymptomatic and diagnosed accidentally (coincidentally) on panoramic radiographs during clinical examination. It was treated with preventive filling of palatal pits. Two dens invaginatus has periodontal problem and the treatment included endododontics.

Large palatocervical groove is a deep groove at the side of the cingulum or on the distal side which may extend up on the root. PMLI with deep palatocervical groove might be susceptible to caries and periodontal disease due to communication with the oral cavity and if severe, communication with the pulp and may require endodontic treatment. In the present study, deep palatocervical grooves were presented in nine subject who needed restorative dentistry and also treatment by a periodontist.

The clinical examination reveals a variety of aesthetic problems such as unpleasant space between teeth as well as deviation of the midline to the affected site in unilateral hypodontia. The smile is an important feature in daily life and should be of interest to general dentists and orthodontists. It is an essential asset for psychosocial adaptation, people with beautiful teeth and smiles are considered more attractive, more intelligent and more popular with the opposite gender [42,43].

In the present study, in 67 (18%) subjects, additional dental specialists (restorative dentist, periodontist, prosthodontist, oral surgeon and endodontist) were included. Among the specialist involved, orthodontists have a crucial role in developing the treatment plan because they have to solve the dilemma concerning which approach should be followed.

Limitations: our study presents some limitations. The study was limited to Caucasian orthodontic populations. The study was retrospective and was based on a convenience sample thus it may not be representative of the population. Although this study may not represent the Slovenian population as a whole, the results are useful for dentist because the patients studied represent the range of dental patient treated due to orthodontic malocclusion and/or dissatisfaction with aesthetics and function.

## 5. Conclusions

Within the limitation of the current study, it was concluded that in orthodontically treated patients:The shape of the PMLI is highly variable and most often occurs as trapezoidal-shaped (59.8%).11.9% of the PMLI had dental developmental anomalies as hypodontia, talon cusp, dens invaginatus and palatocervical groove. Dental anomalies could be asymptomatic or could be associated with the clinical symptoms. Careful observation and appropriate investigation are required to diagnose the developmental condition. Early detection and prompt intervention can alleviate the symptoms in symptomatic cases, prevent premature tooth loss, enhance aesthetics, and overall improve the well-being of the patients.The results of the present study showed that PMLI have important role in aesthetic and function, and often requires interdisciplinary treatment to achieve them.

## Figures and Tables

**Figure 1 diagnostics-12-02759-f001:**
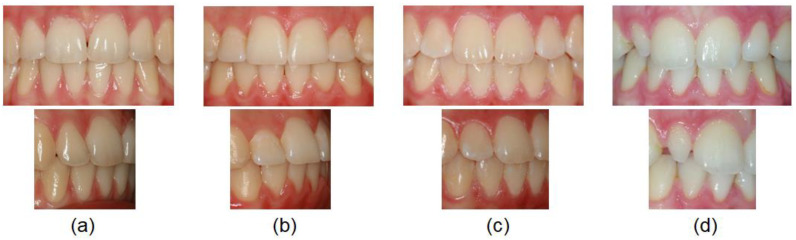
Various crown morphology evaluated as: (**a**) trapezoidal-shaped, (**b**) central incisor-shaped, (**c**) canine-shaped and (**d**) peg-shaped.

**Figure 2 diagnostics-12-02759-f002:**
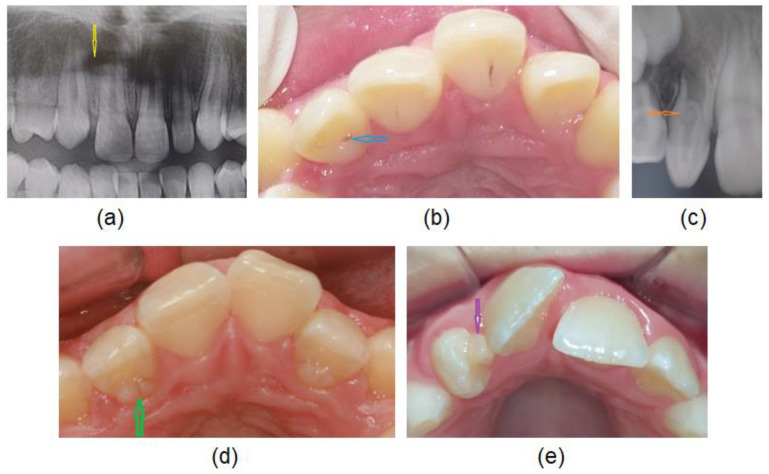
Different dental anomalies: (**a**) hypodontia, (**b**) palatal pit, (**c**) dens invaginatus, (**d**) palatocervical groove and (**e**) talon cusp.

**Figure 3 diagnostics-12-02759-f003:**
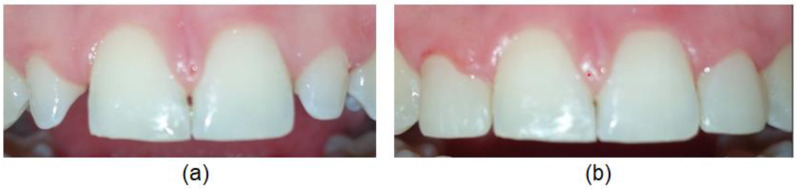
Reshaping the dental crown with composite by a restorative dentist: (**a**) before and (**b**) after buildup.

**Figure 4 diagnostics-12-02759-f004:**
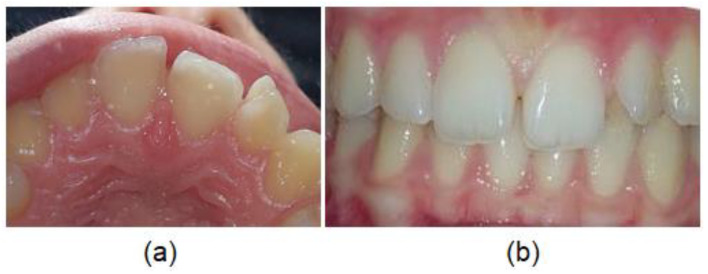
Talon cusp: (**a**) on the palatal surface of the permanent maxillary lateral incisor and (**b**) disturbed occlusion.

**Figure 5 diagnostics-12-02759-f005:**
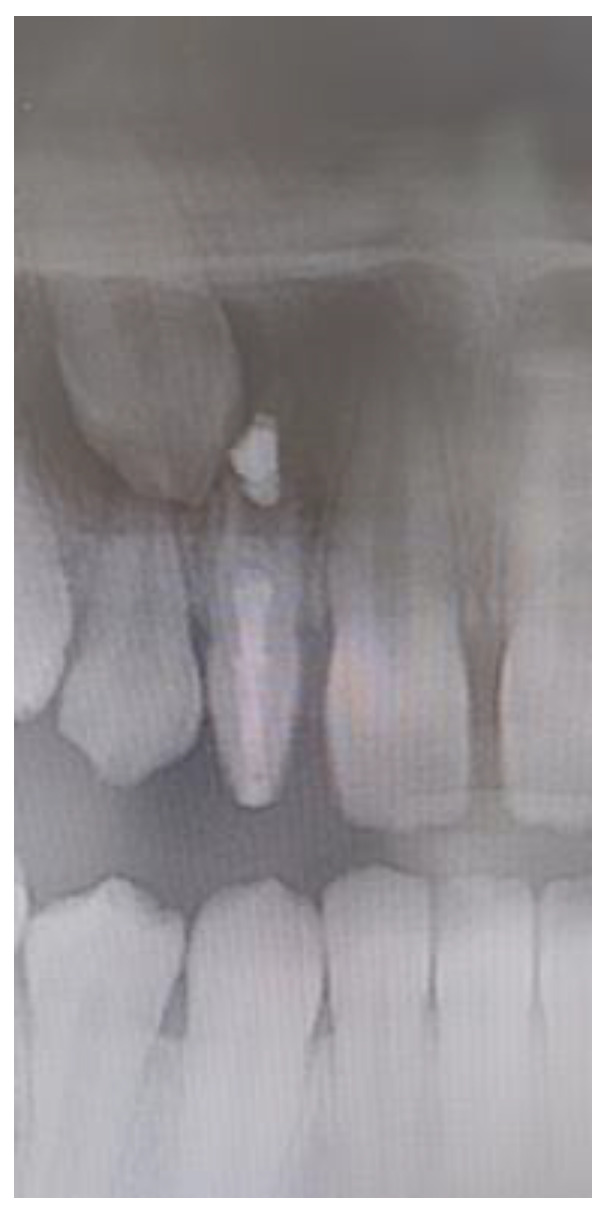
Endodontic treatment of PMLI with dens invaginatus.

**Figure 6 diagnostics-12-02759-f006:**
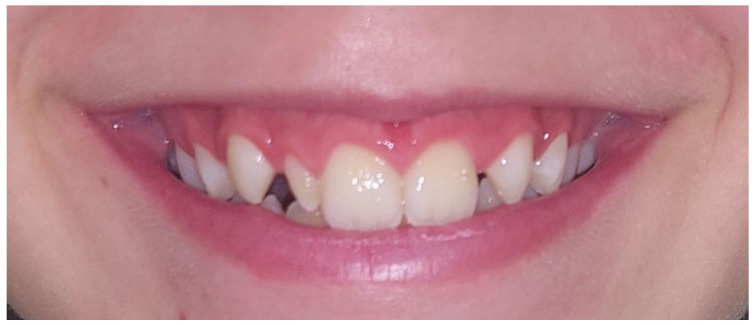
Unattractive smile due to midline asymmetry and space (gaps) caused by right peg-shaped PMLI and hypodontia of left PMLI.

**Table 1 diagnostics-12-02759-t001:** The distribution of various crown shape of permanent maxillary lateral incisors by sex.

	Shape	Trapezoidal-Shaped n (%)	Central Incisor-Shaped n (%)	Canine-Shaped n (%)	Peg-Shaped n (%)	Total n (%)
Sex						
Male		157 (21.7%)	61 (8.4%)	20 (2.8%)	5 (0.7%)	243 (33.6%)
Female		275 (38.1%)	132 (18.3%)	61 (8.4%)	12 (1.6%)	480 (66.4%)
Total		432 (59.8%)	193 (26.7%)	81 (11.2%)	17 (2.3%)	723 (100%)

**Table 2 diagnostics-12-02759-t002:** The distribution of various dental anomalies of permanent maxillary lateral incisors among males and females.

Dental Anomaly	Total Sample n (%)	Sex	Affected Sample n	Affected Teeth n
Hypodontia	15 (4.03%)	M	7	9
		F	8	12
Palatal pit	24 (6.5%)	M	9	14
		F	15	28
Talon cusp	5 (1.3%)	M	2	3
		F	3	4
Dens invaginatus	4 (1.1%)	M	1	1
		F	3	3
Palatocervical groove	9 (2.4%)	M	3	4
		F	6	8

Legends: M: male; F: female.

**Table 3 diagnostics-12-02759-t003:** The distribution of additional specialists included in subjects’ treatment according to crown shapes and affiliated anomalies.

	Specialists	n (%)	Restorative Dentist n (%)	Periodontist n (%)	Prosthodontist n (%)	Oral Surgeon n (%)	Endodontist n (%)
Anomalies							
Crown shape							
Trapezoidal-shaped		2 (0.9%)	2 (100%)	0 (0)	0 (0)	0 (0)	0 (0)
Central incisor-shaped		0	0 (0)	0 (0)	0 (0)	0 (0)	0 (0)
Canine-shaped		0	6 (50)	0 (0)	0 (0)	0 (0)	0 (0)
Peg-shaped		13 (100%)	10 (76.9%)	0 (0)	2 (15.4%)	1 (7.7%)	0 (0)
Dental anomalies							
Hypodontia		15 (100%)	8 (53.4%)	0 (0)	2 (13.3%)	5 (33.3%)	0 (0)
Palatal pit		24 (79.2%)	17 (70.8)	0 (0)	0 (0)	0 (0)	2 (8.3%)
Talon cusp		5 (100%)	5 (100%)	0 (0)	0 (0)	0 (0)	0 (0)
Dens invaginatus		4 (100%)	2 (50%)	0 (0)	0 (0)	0 (0)	2 (50%)
Palatocervical groove		9 (100%)	0 (0)	9 (100)	0 (0)	0 (0)	0 (0)

## Data Availability

The data presented in this study are available on request from the corresponding author.

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
