# Peer review of "Morphological Diversity of Permanent Maxillary Lateral Incisors and Their Impact on Aesthetics and Function in Orthodontically Treated Patients"

_diagnostics, 2022, doi:10.3390/diagnostics12112759_

Round 1
Reviewer 1 Report
Dear authors,
Please consider some suggestions:
Abstract: In the background section you only state the aim of the study. This section should be improved (use the word "aim" instead of "background" or add some real background information).
Sex and gender are different things.
From a biological point of view, it is sex that may influence dental anatomy and morphology, not gender.
Simply put, sex is a biological trait. Gender is a social construction.
As so I would use the word "sex" instead of "gender".
The study evaluates a sample of an orthodontic population, which as you correctly state it's not representative of the general population. In the discussion, a comparison is made with other studies - are the samples from these studies also made up of orthodontic patients? The similarities/differences between the studies is unclear and may be important for the readers to understand the discussion.
I would add the information that an orthodontic population was evaluated to the abstract, aim and conclusion so it would be clear.
Author Response
Dear Reviewer, I added attachment with all corections
I look forward to hearing from you
Sincerely, Anita Fekonja

Reviewer 2 Report
1. The language should polished extensively, because a lot of grammar mistakes and typos can be found throughout the manuscript.
2. How did the 698 subjects be selected? Why there were 294 subjects be excluded? Almost half of the subjects were excluded! In what kind of population did the subjects be selected? Did the selection be randomized?
3. Since the population of this study is only from orthodontic patients, the title of this manuscript should be more specific, and the population should be noted in the title.
4. The clinical significance should be concluded more clearly.
Author Response
Dear Reviewer,
I added attachment with all corrections
I look forward to hearing from you
Sncerely, Anita Fekonja
